# Alcohol Consumption, High-Density Lipoprotein Particles and Subspecies, and Risk of Cardiovascular Disease: Findings from the PREVEND Prospective Study

**DOI:** 10.3390/ijms25042290

**Published:** 2024-02-14

**Authors:** Setor K. Kunutsor, Atanu Bhattacharjee, Margery A. Connelly, Stephan J. L. Bakker, Robin P. F. Dullaart

**Affiliations:** 1Leicester Real World Evidence Unit, Diabetes Research Centre, University of Leicester, Leicester LE5 4WP, UK; 2Division of Population Health and Genomics, University of Dundee, Dundee DD1 4HN, UK; atanustat@gmail.com; 3Labcorp, Morrisville, NC 27560, USA; connem5@labcorp.com; 4Division of Nephrology, Department of Internal Medicine, University Medical Center Groningen, University of Groningen, P.O. Box 30001, 9700 RB Groningen, The Netherlands; s.j.l.bakker@umcg.nl; 5Division of Endocrinology, Department of Internal Medicine, University Medical Center Groningen, University of Groningen, P.O. Box 30001, 9700 RB Groningen, The Netherlands; dull.fam@12move.nl

**Keywords:** HDL cholesterol, HDL particles, HDL subspecies, alcohol consumption, cardiovascular disease, cohort study

## Abstract

The associations of HDL particle (HDL-P) and subspecies concentrations with alcohol consumption are unclear. We aimed to evaluate the interplay between alcohol consumption, HDL parameters and cardiovascular disease (CVD) risk. In the PREVEND study of 5151 participants (mean age, 53 years; 47.5% males), self-reported alcohol consumption and HDL-P and subspecies (small, medium, and large) by nuclear magnetic resonance spectroscopy were assessed. Hazard ratios (HRs) with 95% CIs for first CVD events were estimated. In multivariable linear regression analyses, increasing alcohol consumption increased HDL-C, HDL-P, large and medium HDL, HDL size, and HDL subspecies (H3P, H4P, H6 and H7) in a dose-dependent manner. During a median follow-up of 8.3 years, 323 first CVD events were recorded. Compared with abstainers, the multivariable adjusted HRs (95% CIs) of CVD for occasional to light, moderate, and heavy alcohol consumers were 0.72 (0.55–0.94), 0.74 (0.54–1.02), and 0.65 (0.38–1.09), respectively. These associations remained consistent on additional adjustment for each HDL parameter. For CVD, only HDL-C was associated with a statistically significant decreased risk of CVD in a fully adjusted analysis (HR 0.84, 95% CI 0.72–0.97 per 1 SD increment). For coronary heart disease, HDL-C, HDL-P, medium HDL, HDL size, and H4P showed inverse associations, whereas HDL-C and HDL size modestly increased stroke risk. Except for H6P, alcohol consumption did not modify the associations between HDL parameters and CVD risk. The addition of HDL-C, HDL size, or H4P to a CVD risk prediction model containing established risk factors improved risk discrimination. Increasing alcohol consumption is associated with increased HDL-C, HDL-P, large and medium HDL, HDL size, and some HDL subspecies. Associations of alcohol consumption with CVD are largely independent of HDL parameters. The associations of HDL parameters with incident CVD are generally not attenuated or modified by alcohol consumption.

## 1. Introduction

Cardiovascular disease (CVD) remains a leading cause of morbidity and mortality worldwide, imposing a significant burden on economies and health systems [1]. Established risk factors for CVD include hypertension, diabetes, smoking, obesity, and dyslipidemia [2]. The epidemiology of CVD underscores its deleterious impact across various populations, necessitating a deeper understanding of its risk factors and potential interventions.

A major focus in CVD research has been on high-density lipoprotein cholesterol (HDL-C). Historically, HDL-C has been considered a protective factor due to its role in reverse cholesterol transport—the process which involves transport of excess cholesterol in cells in the periphery to the liver, with ultimate excretion from the body in the feces [3]. Globally, it is acknowledged that HDL-C serves as a consistent and independent predictor of both fatal and non-fatal CVD events, characterized by a strong, graded, and inverse relationship [4,5]. However, emerging evidence suggests a more complex picture [6,7,8]. High levels of HDL-C, paradoxically, have been associated with increased cardiovascular risk [9,10,11,12]. This might be attributable to (i) dysfunctional HDL in some individuals due to elevated levels, potentially increasing cardiovascular risk and (ii) genetic mutations leading to high HDL levels, which might also enhance vascular risk [13]. The conflicting associations observed for HDL-C and CVD may also reflect the fact that HDL does not represent a single uniform lipoprotein fraction (commonly assessed by the HDL-C concentration), but comprises particles that are heterogeneous, vary in composition, size, and function [14], and may have differential associations with CVD [15]. Recent developments in nuclear magnetic resonance (NMR) technology have provided insights into large, medium, and small HDL particles, and enabled the categorization of HDL particles more specifically [16,17,18], thereby facilitating detailed understanding of their relationships with cardiometabolic outcomes in large-scale studies. Indeed, it has been shown that HDL particles and subspecies vary in their associations with incident CVD and type 2 diabetes (T2D) [15,19,20]. For instance, while higher levels of small and medium HDL subspecies (H1, H2 and H4) have been shown to be associated with lower risk of cardiovascular endpoints, no associations have been demonstrated for the large subspecies (H6–H7) [15]. Identifying the specific HDL subspecies that drive the associations of HDL-C with adverse cardiovascular outcomes could help in the development of potential therapeutic targets. The interplay between HDL and alcohol consumption presents an additional layer of complexity. The overall impact of alcohol on cardiovascular health is complex and multifaceted, as there are both benefits and risks associated with alcohol consumption. There is inconsistency regarding the impact of alcohol consumption on different cardiovascular outcomes and uncertainty about alcohol consumption thresholds associated with the lowest risk of adverse cardiovascular outcomes [21,22,23,24]. Furthermore, a number of studies have shown that moderate alcohol consumption is associated with a higher risk of some cardiovascular endpoints [24,25], thereby challenging the concept that moderate alcohol consumption is universally associated with lower cardiovascular risk. Alcohol consumption is generally appreciated to elevate HDL-C levels and has been partly credited for the inverse association between moderate alcohol intake and CVD [26,27]. Consistently, the inverse association of alcohol consumption with myocardial infarction (MI) has been observed to be attenuated following adjustment for HDL-C [21]. In this context, the current study was initiated with two main aims. The first aim was to thoroughly investigate the associations of various HDL parameters—including HDL-C, HDL particle concentration (HDL-P), different HDL subspecies, and HDL size—with self-reported alcohol consumption. The second aim was to examine the impact of various HDL parameters on incident CVD taking account of alcohol consumption both as a confounder and an effect modifier in these associations, thereby providing novel insights into the complex dynamics between HDL, alcohol intake, and cardiovascular risk. In a subsidiary analysis, we also assessed the extent to which measurements of HDL parameters could improve the prediction of CVD.

## 2. Results

### 2.1. Baseline Characteristics

Baseline descriptive characteristics of the participants overall and by categories of self-reported alcohol consumption are shown in Table 1. The mean age of participants at study entry was 53 (SD 12) years and 47.5% were males. Heavy consumers of alcohol (>30 g/d) compared to other categories of alcohol consumption were more likely to be male and current smokers, less likely to use lipid-lowering medication, have higher levels of blood pressure, hsCRP, eGFR, GGT, ALT, AST, total cholesterol, triglycerides, HDL-C, HDL-P, medium HDL, small HDL, H2P, H3P, and H4P, and have lower levels of large HDL, HDL size, H1P, H5P, and H6P. Alcohol abstainers (never/rarely) were older, had higher BMI, FPG, cystatin C, had lower albumin levels and H7P, and were more likely to have prevalent T2D and use antihypertensive medication.

Cross-sectional correlations showed weak to strong positive correlations of HDL-C with other HDL parameters: HDL-P (r = 0.61), large HDL (r = 0.79), medium HDL (r = 0.54), HDL size (r = 0.78), H3P (r = 0.33), H4P (r = 0.51), H5P (r = 0.13), H6P (r = 0.59), and H7P (r = 0.60); there were inverse correlations for small HDL (r = −0.16), H1P (r = −0.09) and H2P (r = −0.12). Large HDL correlated positively with medium HDL (r = 0.23), but inversely with small HDL (r = −0.33). Medium HDL was inversely correlated with small HDL (r = −0.47) (Figure 1).

### 2.2. Relationships between Alcohol Consumption and HDL Parameters

In multivariable linear regression analysis adjusted for age, sex, smoking status, history of T2D, systolic blood pressure (SBP), total cholesterol, albumin, GGT, and ALT, increasing alcohol consumption was associated with higher levels of HDL-C, HDL-P, large HDL, medium HDL, and greater HDL size in a dose-dependent manner (Table 2). Regarding the small HDL subspecies (H1P–H2P), increasing alcohol consumption was associated with decreased levels of H1P, but increased levels of H2P. Both HDL subspecies comprising medium HDL (H3P–H4P) were positively associated with increasing alcohol consumption. Regarding large HDL subspecies (H5P–H7P), increasing alcohol consumption was associated with increased levels of H6P and H7P, but decreased levels of H5P (Table 2).

### 2.3. Relationships between Alcohol Consumption, Plasma Lipid Transfer Proteins, LCAT and Apo A-I

We had access to additional measures of lipid transfer proteins, LCAT activity, and ApoA-I in a subsample of 165 participants. These measures included plasma CETP mass, LCAT activity, PLTP activity, and Apo A-I. In unadjusted analysis, increasing alcohol consumption at the level of >30 g alcohol/day was significantly associated with increased levels of LCAT activity and Apo A-I (Appendix A). In multivariable linear regression analysis adjusted for age, sex, smoking status, history of T2D, SBP, total cholesterol, albumin, GGT, and ALT, alcohol consumption was not significantly associated with levels of CETP mass, LCAT activity, and PLTP activity. However, increasing alcohol consumption at the level of >30 g alcohol/day was significantly associated with increased levels of Apo A-I (Appendix A).

### 2.4. Self-Reported Alcohol Consumption and Incident CVD

During a median follow-up of 8.3 (IQR, 7.7–8.9) years, corresponding to 40,380 person years at risk, 323 incident CVD events (annual rate 8.00/1000 person years at risk; 95% CI: 7.17–8.92) were recorded. There were 232 CHD events and 87 stroke events. Table 3 shows the associations of alcohol consumption assessed by self-reports with the risk of composite CVD. Compared with abstainers, the crude HRs (95% CIs) of CVD for 0.1–10 g/d, 10–30 g/d and >30 g/d were 0.64 (0.49–0.82), 0.77 (0.57–1.04), and 1.00 (0.61–1.65), respectively. On adjustment for age, sex, smoking status, history of T2D, SBP, total cholesterol, triglycerides, BMI, glucose, eGFR, hsCRP, albumin, GGT, and ALT, the corresponding HRs (95% CIs) were 0.72 (0.55–0.94), 0.74 (0.54–1.02), and 0.65 (0.38–1.09), respectively. The associations remained consistent on additional adjustment for each HDL parameter (Table 3). The associations of alcohol consumption categories with CHD were qualitatively similar to that of CVD but did not retain statistical significance (Table 3).

### 2.5. HDL Parameters and Incident CVD

Table 4 and Figure 2 show the associations of HDL parameters with the risk of cardiovascular outcomes. For the composite CVD outcome, only HDL-C was associated with a statistically significant decreased risk of CVD in analysis adjusted for age, sex, smoking status, history of T2D, SBP, total cholesterol, triglycerides, BMI, glucose, eGFR, hsCRP, albumin, GGT, and ALT: HR (95% CI) of 0.82 (0.71–0.95) per 1 SD increment in HDL-C, which remained consistent on further adjustment for alcohol consumption (HR 0.84, 95% CI 0.72–0.97). The minimally adjusted associations of HDL-P, large and medium HDL, HDL size, and H6P were more or less similar compared to that of HDL-C, but lost significance after adjustment for triglycerides, BMI, glucose, eGFR, hsCRP, albumin, GGT, and ALT. There were no associations with small HDL, H1P, H2P, and H5P.

Except for HDL-P, associations of all other HDL parameters with risk of composite CVD were similar in males and females (*p*-value for interaction for all >0.05). HDL-P was associated with a statistically significant decreased risk of CVD in females (HR = 0.79, 95% CI: 0.63–0.99) with no evidence of an association in males (HR = 0.98, 95% CI: 0.84–1.14) (*p*-value for interaction = 0.054) (Appendix A).

For the endpoint of CHD, HDL-C, HDL-P, medium HDL, HDL size, and H4P showed inverse associations with CHD in analyses adjusted for age, sex, smoking status, history of diabetes, SBP, total cholesterol, triglycerides, BMI, glucose, eGFR and hsCRP, albumin, GGT, and ALT: HRs (95% CIs) of 0.70 (0.58–0.85), 0.86 (0.74–0.99), 0.82 (0.69–0.96), 0.78 (0.64–0.94), and 0.81 (0.68–0.97), respectively, per 1 SD increase in these parameters (Table 4). The associations remained consistent on additional adjustment for alcohol consumption. For stroke, there was modest evidence of associations between increased levels of HDL-C and HDL size and an increased stroke risk in fully adjusted analysis: HRs (95% CIs) of 1.26 (0.99–1.60) and 1.29 (0.99–1.68), respectively, per 1 SD increase in these parameters (Table 4). 

Appendix A show the multivariable adjusted associations of HDL parameters with the risk cardiovascular outcomes with further adjustment for HDL-C or HDL-P. The associations of HDL-P, medium HDL, HDL size, and H4P with CHD risk were completely attenuated on additional adjustment for HDL-C (Appendix A). On the contrary, these associations remained consistent on additional adjustment for HDL-P (Appendix A). 

In interaction analyses, except for H6P, the associations of all HDL parameters with the risk of CVD were not significantly modified by alcohol consumption (*p*-value for interaction for all >0.05). Increased H6P levels were associated with a decreased risk of CVD in heavy consumers of alcohol (>30 g/d) (HR = 0.29, 95% CI, 0.09–0.96) but not in other alcohol consumption categories (*p*-value for interaction = 0.042) (Appendix A).

**Table 3 ijms-25-02290-t003:** Associations of alcohol consumption with cardiovascular disease.

Model	0/Rarely	0.1–10 g/Day		10–30 g/Day		>30 g/Day	
		HR (95% CI)	*p*-Value	HR (95% CI)	*p*-Value	HR (95% CI)	*p*-Value
Cardiovascular Disease	N = 1262; 101 Events	N = 2518; 131 Events		N = 1151; 73 Events		N = 220; 18 Events	
Unadjusted model	ref	0.64 (0.49–0.82)	0.001	0.77 (0.57–1.04)	0.089	1.00 (0.61–1.65)	0.99
Base model	ref	0.72 (0.55–0.94)	0.017	0.74 (0.54–1.02)	0.066	0.65 (0.38–1.09)	0.10
Base model plus HDL-C	ref	0.74 (0.57–0.97)	0.028	0.79 (0.57–1.09)	0.15	0.71 (0.42–1.22)	0.22
Base model plus HDL-P	ref	0.73 (0.56–0.96)	0.022	0.77 (0.56–1.06)	0.11	0.68 (0.40–1.15)	0.15
Base model plus large HDL	ref	0.73 (0.56–0.95)	0.019	0.75 (0.55–1.03)	0.076	0.66 (0.39–1.11)	0.12
Base model plus medium HDL	ref	0.73 (0.56–0.95)	0.021	0.77 (0.56–1.06)	0.11	0.68 (0.40–1.16)	0.16
Base model plus small HDL	ref	0.72 (0.55–0.94)	0.017	0.74 (0.54–1.02)	0.066	0.65 (0.38–1.09)	0.10
Base model plus HDL size	ref	0.73 (0.56–0.95)	0.020	0.75 (0.55–1.04)	0.081	0.67 (0.39–1.14)	0.14
Base model plus H1P	ref	0.72 (0.55–0.94)	0.017	0.74 (0.54–1.02)	0.069	0.65 (0.38–1.10)	0.11
Base model plus H2P	ref	0.72 (0.55–0.94)	0.017	0.74 (0.54–1.02)	0.067	0.64 (0.38–1.09)	0.10
Base model plus H3P	ref	0.73 (0.56–0.95)	0.019	0.76 (0.55–1.04)	0.090	0.67 (0.39–1.13)	0.13
Base model plus H4P	ref	0.73 (0.56–0.95)	0.020	0.75 (0.55–1.03)	0.079	0.66 (0.39–1.13)	0.13
Base model plus H5P	ref	0.74 (0.55–0.99)	0.041	0.80 (0.56–1.13)	0.20	0.70 (0.39–1.24)	0.22
Base model plus H6P	ref	0.73 (0.55–0.97)	0.028	0.83 (0.60–1.15)	0.27	0.65 (0.37–1.14)	0.14
Base model plus H7P	ref	0.70 (0.53–0.93)	0.014	0.72 (0.51–1.01)	0.055	0.71 (0.41–1.23)	0.23
**Coronary heart disease**	**N = 1262; 70 events**	**N = 2518; 97 events**		**N = 1151; 54 events**		**N = 220; 11 events**	
Unadjusted model	ref	0.68 (0.50–0.93)	0.015	0.82 (0.58–1.17)	0.28	0.88 (0.47–1.66)	0.70
Base model	ref	0.75 (0.55–1.03)	0.075	0.75 (0.52–1.09)	0.13	0.54 (0.28–1.04)	0.067
Base model plus HDL-C	ref	0.78 (0.57–1.07)	0.13	0.83 (0.57–1.21)	0.33	0.63 (0.32–1.23)	0.17
Base model plus HDL-P	ref	0.76 (0.56–1.05)	0.096	0.79 (0.54–1.15)	0.22	0.58 (0.30–1.13)	0.11
Base model plus large HDL	ref	0.75 (0.55–1.04)	0.082	0.76 (0.52–1.11)	0.15	0.55 (0.28–1.08)	0.081
Base model plus medium HDL	ref	0.76 (0.56–1.05)	0.094	0.79 (0.54–1.15)	0.23	0.59 (0.30–1.15)	0.12
Base model plus small HDL	ref	0.75 (0.54–1.03)	0.073	0.75 (0.51–1.09)	0.13	0.54 (0.28–1.04)	0.066
Base model plus HDL size	ref	0.76 (0.55–1.04)	0.088	0.77 (0.53–1.12)	0.17	0.57 (0.29–1.12)	0.10
Base model plus H1P	ref	0.75 (0.55–1.03)	0.074	0.75 (0.51–1.09)	0.13	0.53 (0.27–1.04)	0.064
Base model plus H2P	ref	0.75 (0.54–1.02)	0.070	0.74 (0.51–1.07)	0.11	0.52 (0.27–1.02)	0.057
Base model plus H3P	ref	0.76 (0.55–1.04)	0.083	0.77 (0.53–1.13)	0.18	0.56 (0.29–1.10)	0.091
Base model plus H4P	ref	0.76 (0.55–1.04)	0.084	0.76 (0.53–1.11)	0.16	0.57 (0.29–1.10)	0.093
Base model plus H5P	ref	0.79 (0.56–1.12)	0.19	0.92 (0.61–1.38)	0.69	0.65 (0.32–1.32)	0.24
Base model plus H6P	ref	0.73 (0.52–1.02)	0.063	0.83 (0.57–1.22)	0.35	0.54 (0.27–1.08)	0.083
Base model plus H7P	ref	0.69 (0.49–0.97)	0.030	0.69 (0.46–1.03)	0.072	0.61 (0.31–1.19)	0.15

1P–H7P, high-density lipoprotein 1–7 particles (from smallest to largest); HDL-C, high-density lipoprotein cholesterol; HDL-P, high-density lipoprotein particles. The analyses are based on 5151 participants (without pre-existing cardiovascular disease, CVD), 323 CVD and 232 coronary heart disease events. Base model includes adjustment for age, sex, smoking status, history of type 2 diabetes, systolic blood pressure, total cholesterol, triglycerides, body mass index, glucose, estimated glomerular filtration rate, high-sensitivity C-reactive protein, albumin, gamma glutamyltransferase, and alanine aminotransferase; ref, denotes the reference category used for comparison.

**Table 4 ijms-25-02290-t004:** Associations of HDL particles and subspecies with cardiovascular outcomes.

HDL Exposures per SD Increase	Model 1		Model 2		Model 3		Model 4	
	HR (95% CI)	*p*-Value	HR (95% CI)	*p*-Value	HR (95% CI)	*p*-Value	HR (95% CI)	*p*-Value
**Cardiovascular Disease**								
HDL-C	0.74 (0.65–0.85)	<0.001	0.77 (0.67–0.88)	<0.001	0.82 (0.71–0.95)	0.009	0.84 (0.72–0.97)	0.021
HDL-P	0.87 (0.77–0.97)	0.013	0.83 (0.74–0.93)	0.001	0.90 (0.80–1.02)	0.098	0.92 (0.81–1.04)	0.17
Large HDL	0.77 (0.66–0.89)	<0.001	0.84 (0.73–0.98)	0.023	0.92 (0.78–1.08)	0.30	0.93 (0.79–1.09)	0.37
Medium HDL	0.80 (0.70–0.91)	0.001	0.85 (0.75–0.96)	0.010	0.88 (0.78–1.01)	0.070	0.90 (0.79–1.03)	0.12
Small HDL	1.09 (0.97–1.23)	0.14	0.98 (0.86–1.11)	0.73	1.01 (0.89–1.15)	0.88	1.01 (0.89–1.15)	0.83
HDL size	0.76 (0.66–0.87)	<0.001	0.84 (0.74–0.97)	0.015	0.89 (0.76–1.04)	0.14	0.90 (0.77–1.06)	0.20
H1P	0.98 (0.88–1.10)	0.78	0.97 (0.87–1.08)	0.63	1.02 (0.91–1.14)	0.74	1.01 (0.90–1.13)	0.84
H2P	1.11 (1.00–1.25)	0.060	1.00 (0.89–1.13)	0.99	0.99 (0.87–1.13)	0.92	1.00 (0.88–1.14)	0.94
H3P	0.85 (0.75–0.96)	0.009	0.89 (0.79–1.01)	0.061	0.92 (0.81–1.04)	0.19	0.93 (0.82–1.05)	0.26
H4P	0.86 (0.76–0.99)	0.032	0.88 (0.77–1.00)	0.050	0.91 (0.80–1.04)	0.19	0.92 (0.81–1.06)	0.25
H5P	0.91 (0.82–1.02)	0.10	0.93 (0.83–1.04)	0.19	0.94 (0.83–1.05)	0.27	0.93 (0.83–1.04)	0.22
H6P	0.84 (0.75–0.93)	0.002	0.88 (0.78–0.99)	0.033	0.93 (0.82–1.06)	0.28	0.94 (0.83–1.07)	0.33
H7P	0.87 (0.78–0.97)	0.012	0.90 (0.80–1.00)	0.059	0.94 (0.83–1.06)	0.30	0.95 (0.84–1.07)	0.40
**Coronary heart disease**								
HDL-C	0.66 (0.56–0.78)	<0.001	0.67 (0.56–0.79)	<0.001	0.70 (0.58–0.85)	<0.001	0.72 (0.59–0.87)	0.001
HDL-P	0.84 (0.74–0.96)	0.013	0.80 (0.70–0.91)	0.001	0.86 (0.74–0.99)	0.040	0.87 (0.75–1.01)	0.076
Large HDL	0.68 (0.56–0.82)	<0.001	0.75 (0.63–0.91)	0.003	0.83 (0.68–1.02)	0.081	0.84 (0.68–1.04)	0.10
Medium HDL	0.72 (0.61–0.84)	<0.001	0.77 (0.66–0.89)	0.001	0.82 (0.69–0.96)	0.014	0.83 (0.71–0.98)	0.026
Small HDL	1.17 (1.02–1.34)	0.027	1.04 (0.90–1.20)	0.63	1.04 (0.89–1.21)	0.62	1.04 (0.90–1.22)	0.59
HDL size	0.65 (0.55–0.77)	<0.001	0.73 (0.62–0.87)	<0.001	0.78 (0.64–0.94)	0.011	0.79 (0.65–0.96)	0.018
H1P	0.96 (0.84–1.09)	0.50	0.95 (0.83–1.07)	0.38	0.99 (0.87–1.13)	0.91	0.98 (0.86–1.12)	0.77
H2P	1.22 (1.07–1.39)	0.002	1.09 (0.95–1.25)	0.21	1.05 (0.91–1.23)	0.49	1.07 (0.92–1.25)	0.37
H3P	0.81 (0.70–0.94)	0.006	0.86 (0.74–0.99)	0.035	0.89 (0.77–1.03)	0.12	0.91 (0.78–1.05)	0.18
H4P	0.74 (0.63–0.88)	<0.001	0.77 (0.65–0.90)	0.001	0.81 (0.68–0.97)	0.021	0.82 (0.69–0.98)	0.029
H5P	0.92 (0.81–1.05)	0.23	0.95 (0.83–1.08)	0.42	0.96 (0.84–1.11)	0.59	0.95 (0.83–1.10)	0.51
H6P	0.82 (0.72–0.93)	0.003	0.86 (0.76–0.99)	0.035	0.91 (0.79–1.06)	0.24	0.92 (0.79–1.07)	0.28
H7P	0.82 (0.72–0.92)	0.001	0.84 (0.74–0.96)	0.008	0.88 (0.77–1.02)	0.095	0.90 (0.77–1.03)	0.13
**Stroke**								
HDL-C	1.06 (0.84–1.33)	0.64	1.12 (0.89–1.41)	0.33	1.22 (0.97–1.54)	0.094	1.26 (0.99–1.60)	0.056
HDL-P	1.01 (0.82–1.26)	0.90	1.01 (0.81–1.26)	0.93	1.11 (0.88–1.39)	0.39	1.13 (0.90–1.44)	0.30
Large HDL	1.01 (0.80–1.28)	0.94	1.09 (0.86–1.38)	0.47	1.21 (0.94–1.55)	0.14	1.22 (0.95–1.56)	0.12
Medium HDL	1.12 (0.89–1.42)	0.32	1.16 (0.93–1.45)	0.19	1.18 (0.94–1.48)	0.16	1.20 (0.95–1.51)	0.13
Small HDL	0.93 (0.74–1.17)	0.53	0.86 (0.67–1.09)	0.20	0.90 (0.70–1.15)	0.38	0.90 (0.70–1.16)	0.43
HDL size	1.08 (0.85–1.36)	0.54	1.17 (0.93–1.48)	0.19	1.27 (0.98–1.65)	0.071	1.29 (0.99–1.68)	0.056
H1P	0.93 (0.76–1.15)	0.53	0.93 (0.75–1.15)	0.50	0.97 (0.78–1.21)	0.80	0.98 (0.79–1.21)	0.83
H2P	0.97 (0.78–1.21)	0.82	0.90 (0.72–1.14)	0.38	0.91 (0.71–1.15)	0.43	0.91 (0.72–1.16)	0.46
H3P	1.07 (0.85–1.34)	0.56	1.11 (0.90–1.38)	0.32	1.14 (0.91–1.42)	0.26	1.14 (0.91–1.42)	0.25
H4P	1.12 (0.89–1.40)	0.33	1.11 (0.89–1.38)	0.36	1.10 (0.88–1.37)	0.41	1.12 (0.89–1.40)	0.33
H5P	0.90 (0.73–1.10)	0.31	0.91 (0.74–1.12)	0.37	0.92 (0.74–1.14)	0.44	0.92 (0.74–1.14)	0.42
H6P	0.91 (0.72–1.14)	0.41	0.96 (0.76–1.22)	0.77	1.05 (0.82–1.36)	0.69	1.06 (0.82–1.37)	0.67
H7P	1.06 (0.84–1.34)	0.64	1.10 (0.86–1.40)	0.45	1.15 (0.88–1.49)	0.31	1.16 (0.89–1.52)	0.27

The analyses are based on 5151 participants (without pre-existing cardiovascular disease, CVD), 323 CVD, 232 coronary heart disease and 87 stroke events. CI, confidence interval; H1P–H7P, high-density lipoprotein 1–7 particles (from smallest to largest); HDL-C, high-density lipoprotein cholesterol; HDL-P, high-density lipoprotein particles; HR, hazard ratio; SD, standard deviation. Model 1: age and sex. Model 2: Model 1 plus smoking status, history of type 2 diabetes, systolic blood pressure, total cholesterol. Model 3: Model 2 plus triglycerides, body mass index, glucose, estimated glomerular filtration rate (as calculated using the Chronic Kidney Disease Epidemiology Collaboration combined creatinine– cystatin C equation), high-sensitivity C-reactive protein, albumin, gamma glutamyltransferase, and alanine aminotransferase. Model 4: Model 3 plus alcohol consumption.

**Figure 2 ijms-25-02290-f002:**
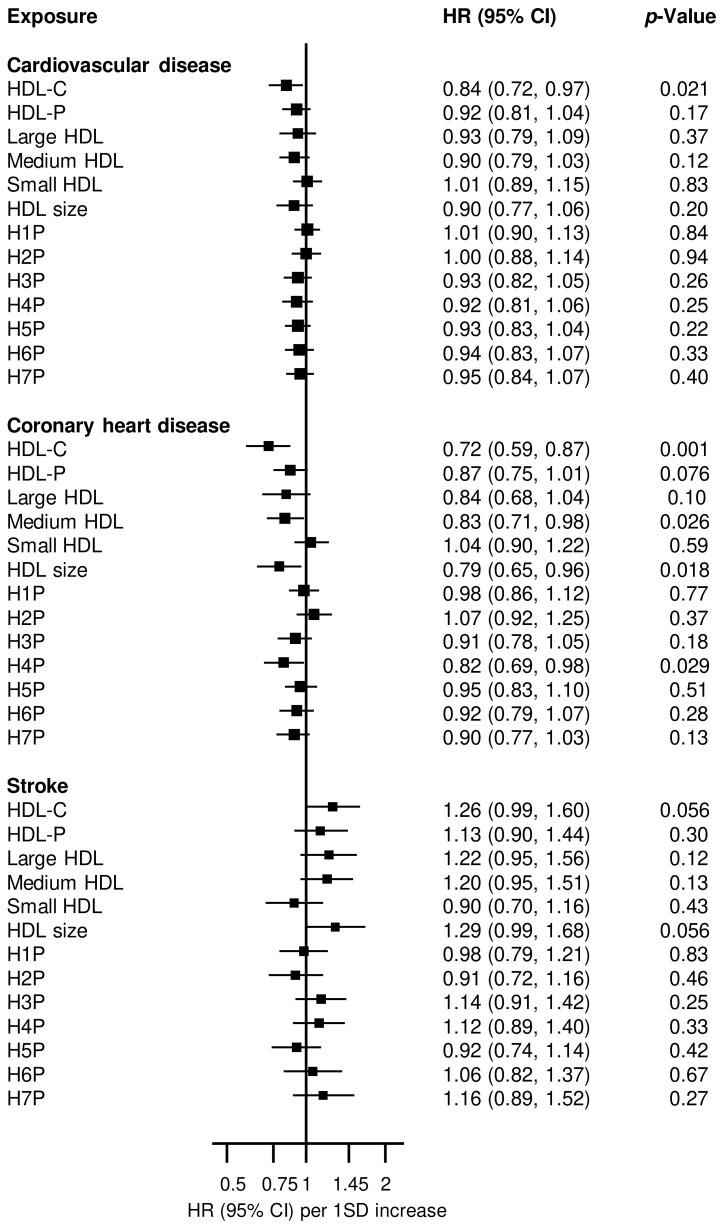
Associations of HDL parameters with cardiovascular outcomes. The analyses are based on 5151 participants (without pre-existing cardiovascular disease, CVD), 323 CVD, 232 coronary heart disease and 87 stroke events. CI, confidence interval; H1P-H7P, high-density lipoprotein 1–7 particles (from smallest to largest); HDL-C, high-density lipoprotein cholesterol; HDL-P, high-density lipoprotein particles; HR, hazard ratio; SD, standard deviation. Analyses are adjusted for age, sex, smoking status, history of type 2 diabetes, systolic blood pressure, total cholesterol, triglycerides, body mass index, glucose, estimated glomerular filtration rate, high-sensitivity C-reactive protein, albumin, gamma glutamyltransferase, alanine aminotransferase, and alcohol consumption.

### 2.6. HDL Parameters and CVD Risk Prediction

A CVD risk prediction model containing traditional risk factors yielded a C-index of 0.7932 (95% CI: 0.7749 to 0.8116). After addition of information on each of the HDL parameters, only HDL-C, HDL size and H4P showed significant changes in the C-index: 0.0096 (0.0020 to 0.0171; *p* = 0.013), 0.0064 (0.0004 to 0.0124; *p* = 0.038), and 0.0051 (0.0000 to 0.0101; *p* = 0.050), respectively (Table 5).

## 3. Discussion

### 3.1. Summary of Main Findings

In this large general population-based prospective study, we sought to evaluate the interplay between HDL parameters, alcohol consumption, and cardiovascular risk. Correlation analyses demonstrated weak to strong positive and inverse correlations of HDL-C with other HDL parameters. Increasing alcohol consumption progressively increased not only levels of HDL-C, but also HDL-P, large and medium HDL, HDL size and HDL subspecies (H3P, H4P, H6, and H7) in dose-dependent fashions, with reciprocal decreases in H1P and H5P. Heavy alcohol consumption increased the levels of Apo A-I, but not other measures of HDL functionality. Occasional to light alcohol consumption was associated with a reduced risk of CVD independently of established cardiovascular risk factors. Notably, this association was independent of HDL-C and other HDL parameters. We could not confirm the association between heavy alcohol consumption and increased risk of CVD [24]. There were varying associations of HDL parameters with cardiovascular risk. For composite CVD, only HDL-C was associated with a decreased risk of CVD; HDL-C, HDL-P, medium HDL, HDL size, and H4P showed inverse associations with CHD risk, whereas increases in the levels of HDL-C and HDL size were each associated with an increased stroke risk. These associations remained consistent on additional adjustment for alcohol consumption. Except for H6P, the associations of all HDL parameters with risk of CVD were unmodified by alcohol consumption; increased H6P levels were associated with a decreased risk of CVD in heavy consumers of alcohol. Except for HDL-P, the associations of HDL parameters with risk of composite CVD were similar in males and females; high HDL-P was associated with a decreased risk of CVD in females with no evidence of an association in males. Notably, the multivariable-adjusted associations of non-HDL-C based HDL particle parameters with cardiovascular outcomes were completely attenuated on additional adjustment for HDL-C, but remained consistent when they were adjusted for HDL-P. Finally, the addition of HDL-C, HDL size or H4P to a CVD risk prediction model containing established risk factors was associated with significant improvements in the discrimination of long-term CVD risk.

### 3.2. Comparison with Previous Work

Analysis from the multiethnic Dallas Heart Study, which was based on the same NMR methodology but used an earlier algorithm to determine lipoprotein subfractions, showed that increasing alcohol consumption coincided with an increase in HDL-P and HDL size besides an increase in HDL-C, similar to the current findings [28]. In a recent pooled multiethnic cohort comprising four studies, namely Atherosclerosis Risk in Communities (ARIC), Dallas Heart Study (DHS), Multi-Ethnic Study of Atherosclerosis (MESA), and PREVEND, Deets and colleagues [15] used exactly the same NMR methodology and algorithm to assess whether levels of specific size-based HDL subspecies were associated with atherosclerotic CVD and improved risk prediction beyond traditional risk factors. Of the 7 HDL subspecies, higher levels of H1P (small) and H4P (medium) were consistently associated with a reduced risk of MI, ischemic stroke, and the composite end point of cardiovascular death, MI, and all stroke, with H4P showing an improvement in risk prediction for MI. Our results showed an inverse association between medium HDL and CHD; we observed no association between small HDL and a cardiovascular endpoint. In addition to H4P, we also showed that HDL size improved CVD risk prediction. In another pooled multiethnic analysis of the ARIC, DHS, MESA, and PREVEND studies, Singh and colleagues [29] compared the associations of HDL-C and HDL-P with ischemic events (MI and ischemic stroke). In that study, an earlier NMR-based algorithm was used to measure HDL-P and HDL subfractions. Their results showed that HDL-P was more consistent than HDL-C in associating with MI and ischemic stroke in the general population and in women. Furthermore, HDL-P attenuated all associations between HDL-C and events, whereas HDL-C had negligible effects on associations between HDL-P and events in the overall population [29]. In contrast, our results in predominantly White individuals showed that HDL-C was a somewhat stronger risk indicator for cardiovascular outcomes than HDL-P; in addition, the associations of HDL parameters with cardiovascular outcomes were completely attenuated on additional adjustment for HDL-C, but remained consistent on additional adjustment for HDL-P. Our key finding of a protective effect of occasional to light alcohol consumption on cardiovascular risk is consistent with previous studies [23,26,27]. The association was also independent of all evaluated HDL parameters including HDL-C, a finding which contrasts with another meta-analysis [21].

### 3.3. Possible Explanations for Findings

Several of our findings deserve explanation. The increase in HDL-C, HDL-P, large and medium HDL, HDL size, and some HDL subspecies with increasing alcohol consumption likely reflects the combined effects of stimulation of Apo-AI synthesis [30], stimulation of the reverse cholesterol transport pathway reflected by an increase in cholesterol efflux capacity [28,31], inhibition of concentration and activity of CETP at high alcohol consumption [32], and enhanced lipoprotein lipase activity [33,34]. These pathways reflect HDL functionality and may potentially account for the protective effect of occasional to light alcohol consumption on cardiovascular risk. It has been reported that 40–60% of the beneficial effect of low to moderate alcohol consumption on the risk of CVD is mediated through an increase in HDL-C alone [35]. Indeed, our subsidiary analysis of a subsample of 165 participants showed that increasing alcohol consumption significantly increased Apo A-I levels, with a non-significant decrease in CETP mass. Apo A-I is a major component of HDL particles, and its synthesis is crucial for HDL formation and function [36,37]. Ethanol may stimulate ApoA-I synthesis in vitro, in particular, in HepG2 cells [37], and controlled sustained alcohol ingestion increases plasma ApoA-I in humans [38,39]. Additionally, increasing amounts of alcohol consumption up to 30 g per day were found to be associated with progressively higher serum activity of the anti-oxidative and anti-inflammatory enzyme, paraoxonase-1 (PON-1), which is predominantly related to HDL-P and large HDL particles [40]. PON-1 could be considered as an indirect metric of HDL’s anti-inflammatory function, determined as the ability of HDL to suppress tumor necrosis factor-α- induced vascular cell adhesion molecule-1 mRNA expression in endothelial cells in vitro [41]. Using this approach, we were able to demonstrate an inverse relationship of this HDL function metric with PON-1 [41].

Furthermore, alcohol’s influence on the CETP-mediated cholesteryl ester transfer process [42] could alter the dynamics of lipid exchange between HDL and other lipoprotein particles, impacting HDL composition and function. Common CETP gene variation may also modulate the effect of alcohol consumption on CVD risk [43]. Besides CETP, LCAT and PLTP are key players in HDL metabolism (via the reverse cholesterol transport pathway) and contribute to increased levels of HDL via HDL maturation and remodeling [44,45]. There have been suggestions that moderate alcohol consumption could potentially play a role in the increased levels of HDL-C via increases in the activity levels of LCAT and PLTP [46]; however, the evidence so far shows no impact of alcohol consumption on these factors [46,47]. Our substudy showed no significant evidence of associations between alcohol consumption and the activities of LCAT and PLTP, though LCAT increased with increasing alcohol consumption in the unadjusted analysis.

It is interesting to note that though there is an established body of evidence on the associations between increased alcohol consumption and increased levels of HDL and its subfractions, the evidence on other lipid parameters and their subfractions (low-density lipoprotein, very low-density lipoprotein, triglycerides, lipoprotein (a)) is sparse and inconsistent—studies have shown no change, decreased levels, as well as increased levels of these parameters [48]. It is also noteworthy that a recent study unexpectedly showed lower levels of HDL-C and ApoA-I as well as lower levels of larger and smaller HDL particles with modest but long-term alcohol (at least 10 years) consumption among middle-aged South Korean women [49]. These contrasting results may be related to the duration of alcohol consumption. In our study, alcohol was assessed using self-reported questionnaires and only covered a time period of 1 month. Long-term alcohol consumption may impair HDL functionality and may reflect evidence that long-term heavy alcohol drinking is associated with an increased risk of adverse cardiovascular outcomes [50]. In our study, the lack of an association between heavy alcohol consumption and CVD risk could be attributed to the lack of statistical power in that category (n = 220 participants, 18 CVD events). Among HDL parameters, only HDL-C was shown to be associated with composite CVD risk; however, HDL-C, HDL-P, and other HDL parameters associated with CHD risk. In addition to the interplay between alcohol consumption, HDL functionality, and CVD risk, the functional properties of HDL have also been proposed to explain the observed atherosclerotic cardiovascular protective effects of HDL [7]. Indeed, several studies have shown HDL-C efflux capacity, a key metric of anti-atherosclerotic HDL function, to be associated with reduced risk of CVD independently of HDL-C [51,52]. The observed differential associations of HDL-C and other HDL parameters with CHD compared to stroke may be attributed to distinct mechanisms in atherosclerotic processes. While both conditions are mediated by atherosclerosis, the specific impact of HDL parameters on endothelial function, inflammation, and thrombosis might vary, leading to differing influences on CHD and stroke risk. The contrasting results for HDL-C and HDL-P between our study and others underscore the complexity of HDL biology and its implications for CVD. While HDL-C quantifies the cholesterol carried by HDL in the blood, HDL-P measures the actual number of HDL particles, regardless of their cholesterol content [53]. HDL-P provides a more comprehensive assessment of HDL’s role in lipid metabolism and cardiovascular risk and therefore may be a better marker of cardiovascular risk than HDL-C. Indeed, multiple studies have reported these observations [54,55]. This is an area that requires further investigation. The interplay between HDL, alcohol consumption, and cardiovascular risk deserves further exploration using mechanistic studies.

### 3.4. Strengths and Limitations

This study stands out as the first prospective examination of the relationship between a detailed array of HDL composition and size measures, HDL particles and subtypes as determined by updated NMR spectroscopy, alcohol consumption, and the risk of combined CVD, including specific endpoints like CHD and stroke. The study’s strengths are its large sample size, prolonged follow-up period, exclusion of subjects with pre-existing CVD to reduce reverse causation bias, and the inclusion of a wide array of cardiovascular risk factors allowing for the adjustment of potential confounders. Nonetheless, certain limitations warrant attention. These encompass typical biases in observational studies, such as the inability to establish causality and possible residual confounding due to inaccuracies in measured and unaccounted confounders. Alcohol consumption categories were self-reported, introducing limitations and the risk of misclassification bias [56]. The relatively low incidence rate of stroke precluded an assessment of alcohol consumption’s impact on this outcome. Furthermore, we were unable to assess the influence of specific types of alcoholic beverages on CVD risk. Notably, there are known inconsistencies regarding the effects of different beverages (wine, beer, spirits) on CVD risk, and the debate whether their protective effects stem from their alcoholic content (ethanol) or non-alcoholic components (mainly polyphenols) [57]. Studies suggest that polyphenols in wine and beer might reduce CVD risk independently of ethanol [57]. We were only able to conduct a sex-specific analysis for the associations between HDL parameters and risk of composite CVD given the relatively low events for other cardiovascular endpoints. While we could not extend this approach to the relationship between alcohol consumption (categorical variable) and CVD risk, sex-specific associations have been investigated in several previous studies [23]. Our analyses were based on single baseline assessments of alcohol intake and HDL parameters, which may not accurately reflect participants’ true long-term “usual” exposures throughout the duration of the study, due to the phenomenon of regression dilution bias. Based on findings of studies that accounted for regression dilution bias by using information on repeat assessments of self-reported alcohol intake [35], evaluations based on single-timepoint data could systematically underestimate the true disease risk linked to these factors. Finally, we did not measure cholesterol efflux capacity or other direct metrics of HDL function in the current study, although LCAT activity has been linked to the antioxidative function of HDL [58].

## 4. Materials and Methods

### 4.1. Study Design and Population

This research adhered to the STROBE (Strengthening the Reporting of Observational Studies in Epidemiology) criteria, specifically designed for observational epidemiology studies (Appendix A) [59]. The subjects of this study were from the Prevention of Renal and Vascular End-Stage Disease (PREVEND) initiative. This cohort study, based in the Netherlands, focused on investigating the progression of urinary albumin excretion and its links to renal diseases and CVDs. The methodology and recruitment for this study have been previously detailed [60,61,62,63]. PREVEND received ethical clearance from the Medical Ethics Committee at the University Medical Center Groningen, under the approval number MEC 96/01/022, and aligned with the principles of the Declaration of Helsinki. Informed consent was obtained in writing from all participants. The PREVEND study population included a diverse group of residents from Groningen, the Netherlands. Initially, the study included 8592 participants who underwent the screening process between 1997 and 1998. The baseline phase for this current analysis included 6894 individuals, aged between 28 to 75 years, who were evaluated between 2001 and 2003. Excluded were those with a previous CVD history. The analytic cohort for this study is comprised of 5151 participants who had complete data on HDL parameters, various confounders, self-reported alcohol consumption, and subsequent cardiovascular events. The process for selecting the analytic sample is depicted in Appendix A.

### 4.2. Assessment of Exposures and Other Risk Markers

In this study, participants were seen twice in outpatient settings, during which we collected initial information on demographic characteristics, lifestyle factors, physical assessments, medical backgrounds, and medication usage. Additional details on medications were obtained from a comprehensive pharmacy dispensing database in Groningen, encompassing drug usage data for about 95% of those in the PREVEND program [64]. Hypertension was identified through self-reports of doctor diagnoses, usage of hypertension drugs, or when blood pressure readings were equal to or exceeded 140/90 mmHg. We averaged the final two blood pressure measurements from each visit for our records. Criteria for identifying type 2 diabetes (T2D) included any of the following: fasting plasma glucose levels ≥ 7.0 mmol/L (126 mg/dL), random plasma glucose readings ≥ 11.1 mmol/L (200 mg/dL), a doctor’s diagnosis of T2D, or the start of treatments to lower glucose, as noted in the central pharmacy records [19]. Body mass index (BMI) was calculated using the standard formula: weight in kilograms divided by height in meters squared. Data on alcohol consumption were self-reported [65], utilizing a structured questionnaire with predefined categories over a one-month period, including options such as (i) none or almost never, (ii) 1–4 units/month, (iii) 2–7 units/week, (iv) 1–3 units/day, and (v) over 3 units/day. These responses were then converted into daily alcohol intake, with one drink in the Netherlands equating to 10 g of alcohol, irrespective of beverage type [40]. We later categorized alcohol consumption levels as (i) none or rare, (ii) 0.1–10 g/day (occasional to light), (iii) 10–30 g/day (moderate), and (iv) over 30 g/day (heavy).

For the purpose of biochemical analysis, venous samples of both plasma and serum were collected from the study participants after an overnight fast, as detailed in references [66,67,68,69,70]. These plasma samples underwent a centrifugation process at a temperature of 4 °C. Samples treated with EDTA as an anticoagulant were then stored at a temperature of −80 °C until the time of analysis. The evaluation of lipoprotein levels was performed through nuclear magnetic resonance (NMR) spectroscopy, utilizing the Vantera^®^ Clinical NMR Analyzer at Labcorp, Morrisville, NC, with the implementation of the LP4 algorithm on EDTA plasma specimens exactly as described [19]. Measurements of total cholesterol, triglycerides, and HDL-C were conducted employing partial least squares (PLS) regression techniques. The quantification of small, medium, and large HDL particles, along with the HDL subspecies ranging from H1P to H7P, was achieved by analyzing the distinct amplitudes of their lipid methyl group NMR signals. The overall HDL-P was determined by adding together the concentrations of these small, medium, and large HDL particles. The mean HDL size was calculated as follows: we used weighted averages based on the sum of the diameters of the three sizes of HDL particles (small, medium and large), each multiplied by their respective mass percentages [19]. The diameters for these HDL subclasses and subspecies were defined as follows: small HDL ranged from 7.4 to 8.0 nm, medium HDL from 8.1 to 9.5 nm, and large HDL from 9.6 to 13 nm. Specifically, H1P measured at 7.4 nm, H2P at 7.8 nm, H3P at 8.7 nm, H4P at 9.5 nm, H5P at 10.3 nm, H6P at 10.8 nm, and H7P at 12.0 nm. Within these, small HDL consisted of H1 and H2, medium HDL of H3 and H4, and large HDL of H5 to H7.

Plasma cholesteryl ester transfer protein (CETP) mass (in mg/L) was measured with a double-antibody sandwich enzyme-linked immunosorbent assay as described [71]. Le-cithin:cholesterol acyltransferase (LCAT) activity (in nmol cholesterol esterified per milliliter plasma per h) was determined using excess exogenous substrate containing [3H]-cholesterol as previously described [58]. An LCAT activity of 87 nmol cholesterol esterified per milliliter plasma per h corresponds to the activity measured in human pool reference plasma and to 100 arbitrary units in previous publications [67]. With this assay system, LCAT activity is strongly correlated with LCAT mass. Phospholipid transfer protein (PLTP) activity was measured with a liposome vesicles-HDL system and was expressed in AU relative to human pool referent plasma [72]. CETP mass, LCAT activity, and PLTP activity were measured in duplicate. Apolipoprotein A-I (in g/L) was measured by nephelometry using reagents from Dade Behring (BN II, Marburg, Germany) [73]. All intra- and interassay coefficients of variation were less than 7.5%. 

Plasma fasting glucose was measured by dry chemistry (Eastman Kodak, Rochester, NY, USA). High-sensitivity C-reactive protein (hs-CRP) was assayed by nephelometry (Dade Behring Diagnostic, Marburg, Germany). Circulating albumin, and gamma glutamyltransferasse (GGT) activity were measured by an enzymatic colorimetric method (Roche Modular P; Roche Diagnostics, Mannheim, Germany). Alanine aminotransferase (ALT) and aspartate aminotransferase (AST) were measured using the standardized kinetic method with pyridoxal phosphate activation (Roche Modular P; Roche Diagnostics, Mannheim, Germany) according to the recommendations of the International Federation of Clinical Chemistry [74,75]. Serum creatinine was determined by Kodak Ektachem dry chemistry (Eastman Kodak, Rochester, NY, USA) and serum cystatin C level by nephelometry (BN II N) (Dade Behring Diagnostic, Marburg, Germany). In 2001–2003, creatinine assays were non-IDMS calibrated. To allow for the calculation of estimated glomerular filtration rate (eGFR) based on the Chronic Kidney Disease Epidemiology Collaboration (CKD-EPI) combined creatinine-cystatin C equation, we performed re-measurements of approximately 28,000 available plasma samples of the five PREVEND screenings in 2010–2012 using an IDMS traceable enzymatic Roche assay. Estimated glomerular filtration rate was calculated using the CKD-EPI combined creatinine-cystatin C equation [76]. 

### 4.3. Outcome Ascertainment

In this analysis, the primary outcome was the initial occurrence of composite CVD events, with coronary heart disease (CHD) and stroke as secondary outcomes. Cardiovascular outcomes were classified using the International Classification of Diseases, Ninth Revision (ICD-9) codes up to 1 January 2009, and ICD-10 codes thereafter. Hospitalization data for cardiovascular morbidity were obtained from PRISMANT, the national Dutch registry for hospital discharge diagnoses [77]. The first occurrence of composite CVD was characterized as including acute MI, a combined endpoint of acute and subacute ischemic heart disease (IHD), coronary artery bypass grafting (CABG), percutaneous transluminal coronary angioplasty (PTCA), subarachnoid hemorrhage, intracerebral hemorrhage, other forms of intracranial hemorrhage, occlusions or stenoses of precerebral or cerebral arteries, and other vascular interventions such as percutaneous transluminal angioplasty or bypass grafting of peripheral vessels and the aorta. CHD was specifically defined as fatal or nonfatal IHD, fatal or nonfatal MI, CABG, and PTCA. Stroke encompassed subarachnoid hemorrhage, intracerebral hemorrhage, other specified and unspecified intracranial hemorrhages, occlusions and stenoses of precerebral or cerebral arteries, and carotid artery obstruction. 

### 4.4. Statistical Analyses

Skewed variables (e.g., hsCRP, triglycerides, GGT, ALT, AST, H5P, H6P, and H7P) were natural logarithm (log_e_) transformed to achieve approximate normal distributions. Baseline characteristics were presented as means (standard deviation, SD) or median (interquartile range, IQR) for continuous variables and numbers (percentages) for categorical variables. We assessed univariable relationships between alcohol consumption and baseline characteristics using ANOVA (for continuous variables) and chi-square tests (for categorical variables). Multivariable linear regression analyses were used to assess the relationships of alcohol consumption with HDL parameters. Time-to-event Cox proportional hazards models were used to estimate hazard ratios (HRs) with 95% confidence intervals (Cis) for the associations of self-reported alcohol consumption and HDL parameters with the risk of cardiovascular outcomes, after confirmation of no major departure from the proportionality of hazards assumptions [78]. HDL parameters were modelled per 1 SD increment in parameters (log transformed for parameters which were not normally distributed). The adjustment for confounders were based on four progressive models: (Model 1) age and sex; (Model 2) model 1 plus smoking status, history of T2D, systolic blood pressure (SBP), total cholesterol; (Model 3) model 2 plus triglycerides, body mass index (BMI), fasting plasma glucose, eGFR, hsCRP, albumin, GGT, and ALT; and (Model 4) model 3 plus alcohol consumption. We used interaction tests to assess statistical evidence of effect modification by alcohol consumption on the associations of HDL parameters with CVD. To assess whether adding information on each HDL parameter to conventional cardiovascular risk factors is associated with an improvement in the prediction of CVD risk, we calculated measures of discrimination for censored time-to-event data (Harrell’s C-index [79]). To investigate the change in C-index, we added each HDL parameter to a model based on traditional risk factors included in the World Health Organization (WHO) cardiovascular disease risk prediction score (i.e., age, smoking status, SBP, history of diabetes, and total cholesterol). [80] All statistical analyses were conducted using R Statistical Software (version 2.14.0; R Foundation for Statistical Computing, Vienna, Austria) and Stata MP version 16 (Stata Corp, College Station, TX, USA).

## 5. Conclusions

The study reveals a complex relationship between alcohol consumption, HDL parameters, and cardiovascular risk. Moderate alcohol consumption appears to beneficially influence levels of HDL-C, HDL-P, and other HDL parameters and is associated with a reduced risk of CVD, partly independent of other clinical and laboratory covariates. However, the impact of HDL parameters on cardiovascular outcomes, including CHD and stroke risk, varies, suggesting a subtle role of HDL in cardiovascular health. Importantly, the role of HDL-C as a key factor in these associations is highlighted, as HDL-C appears to be a stronger cardiovascular risk indicator than other parameters and adjustments for HDL-C may alter the observed relationships between other HDL parameters and cardiovascular outcomes. Furthermore, the associations of these HDL parameters with cardiovascular outcomes appear not to be attenuated or modified to a considerable extent by alcohol consumption. Finally, among HDL parameters, HDL-C, HDL size and H4P may provide improvements in the prediction of long-term CVD risk.

## Figures and Tables

**Figure 1 ijms-25-02290-f001:**
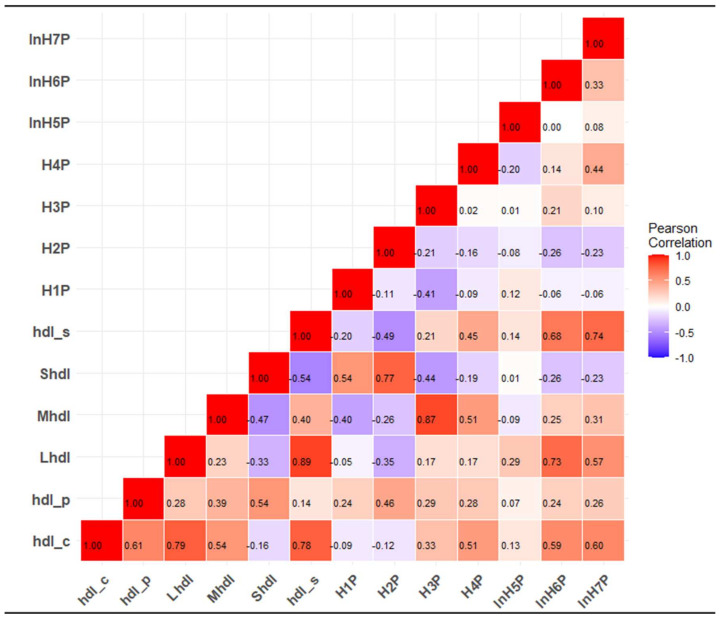
Correlation heat map and matrix for HDL parameters. H1P–H7P, high-density lipoprotein 1–7 particles (from smallest to largest); small HDL: H1P and H2P; medium HDL: H3P–H4P; large HDL: H5P–H7P; hdl_c, high-density lipoprotein cholesterol; hdl_p, high-density lipoprotein particles; Lhdl, large HDL, Mhdl, medium HDL, Shdl, small HDL; significant values are shown in deep colors, with non-significant values shown in light shade.

**Table 1 ijms-25-02290-t001:** Baseline participant characteristics overall and according to alcohol consumption.

		Alcohol Consumption Categories				
Characteristic	Overall (N = 5151)	Never or Rarely (N = 1262)	0.1–10 g/Day (N = 2518)	10–30 g/Day (N = 1151)	>30 g/Day (N = 220)	*p*-Value
	Mean (SD) or Median (IQR)	Mean (SD) or Median (IQR)	Mean (SD) or Median (IQR)	Mean (SD) or Median (IQR)	Mean (SD) or Median (IQR)	
**Questionnaire**						
Male (%)	2445 (47.5%)	419 (33.2%)	1204 (47.8%)	655 (56.9%)	167 (75.9%)	<0.001
Age, years	53 (12)	55 (13)	52 (12)	53 (11)	54 (10)	<0.001
History of diabetes (%)	275 (5.3%)	113 (9.0%)	102 (4.1%)	52 (4.5%)	8 (3.6%)	<0.001
Current smoking (%)	1417 (27.5%)	329 (26.1%)	632 (25.1%)	344 (29.9%)	112 (50.9%)	<0.001
Use of antihypertensive medication (%)	764 (15.8%)	262 (21.7%)	313 (13.4%)	153 (14.0%)	36 (17.2%)	<0.001
Use of lipid lowering medication (%)	118 (2.8%)	37 (3.4%)	48 (2.3%)	33 (3.5%)	0 (0.0%)	0.02
**Physical measurements**						
Body mass index, kg/m^2^	26.6 (4.3)	27.5 (5.1)	26.3 (4.1)	26.0 (3.6)	26.5 (4.3)	<0.001
Waist circumference, cm	91.4 (12.6)	92.5 (13.6)	90.5 (12.3)	91.3 (11.9)	94.7 (12.3)	<0.001
Systolic blood pressure, mmHg	125 (18)	127 (20)	124 (17)	127 (18)	131 (17)	<0.001
Diastolic blood pressure, mmHg	73 (9)	73 (9)	72 (9)	74 (9)	77 (8)	<0.001
**Metabolic, inflammatory, renal, and liver markers**						
Fasting plasma glucose, mmol/L	4.99 (1.11)	5.10 (1.32)	4.93 (1.02)	5.01 (1.09)	5.05 (0.78)	<0.001
High sensitivity CRP, mg/L	1.31 (0.60, 2.95)	1.62 (0.75, 3.53)	1.22 (0.56, 2.81)	1.14 (0.55, 2.53)	1.65 (0.76, 3.57)	<0.001
Creatinine, mg/dL	0.82 (0.20)	0.80 (0.18)	0.82 (0.17)	0.84 (0.28)	0.83 (0.14)	<0.001
Cystatin C, mg/L	0.90 (0.20)	0.93 (0.20)	0.89 (0.17)	0.88 (0.23)	0.90 (0.17)	<0.001
Estimated GFR, mL/min/1.73 m^2^	84.6 (9.8)	82.4 (11.5)	85.2 (9.0)	85.4 (9.5)	86.0 (8.4)	<0.001
Albumin, g/L	44.0 (5.0)	43.4 (2.7)	44.1 (5.8)	44.3 (5.4)	44.2 (2.8)	<0.001
Plasma GGT, U/L	23.0 (15.0, 37.0)	21.0 (15.0, 33.0)	22.0 (15.0, 35.0)	27.0 (17.0, 42.0)	37.0 (25.0, 71.0)	<0.001
Plasma ALT, U/L	17.0 (13.0, 24.0)	16.0 (12.0, 23.0)	17.0 (12.0, 24.0)	18.0 (13.0, 25.0)	20.0 (15.0, 30.0)	<0.001
Plasma AST, U/L	22.0 (19.0, 26.0)	22.0 (19.0, 26.0)	22.0 (19.0, 26.0)	23.0 (20.0, 27.0)	25.0 (21.0, 30.0)	<0.001
**Lipid markers**						
Total cholesterol, mmol/L	5.46 (1.04)	5.42 (1.06)	5.42 (1.03)	5.51 (1.01)	5.78 (1.14)	<0.001
Triglycerides, mmol/L	1.11 (0.80, 1.59)	1.17 (0.85, 1.67)	1.08 (0.78, 1.54)	1.08 (0.80, 1.58)	1.20 (0.85, 1.99)	<0.001
HDL-C, mmol/L	1.27 (0.31)	1.22 (0.29)	1.26 (0.30)	1.33 (0.33)	1.34 (0.36)	<0.001
HDL-P, µmol/L	21.2 (2.7)	20.6 (2.5)	21.1 (2.6)	21.9 (2.9)	22.4 (2.9)	<0.001
Large HDL, µmol/L	1.84 (1.31)	1.79 (1.25)	1.86 (1.30)	1.90 (1.40)	1.64 (1.25)	0.013
Medium HDL, µmol/L	5.23 (2.21)	4.93 (2.03)	5.14 (2.09)	5.59 (2.40)	5.97 (2.89)	<0.001
Small HDL, µmol/L	14.2 (2.9)	13.9 (2.7)	14.1 (2.8)	14.4 (3.1)	14.8 (3.1)	<0.001
HDL size, nm	8.98 (0.46)	8.96 (0.44)	8.98 (0.45)	9.00 (0.48)	8.94 (0.45)	0.17
H1P, µmol/L	3.51 (1.78)	3.43 (1.77)	3.59 (1.79)	3.47 (1.78)	3.18 (1.85)	0.001
H2P, µmol/L	10.7 (2.4)	10.4 (2.3)	10.5 (2.4)	10.9 (2.6)	11.7 (2.6)	<0.001
H3P, µmol/L	3.33 (1.83)	3.15 (1.74)	3.28 (1.75)	3.57 (1.92)	3.74 (2.41)	<0.001
H4P, µmol/L	1.89 (1.10)	1.78 (1.00)	1.86 (1.06)	2.03 (1.20)	2.23 (1.35)	<0.001
H5P, µmol/L	0.29 (0.05, 0.61)	0.34 (0.08, 0.66)	0.30 (0.05, 0.62)	0.26 (0.01, 0.57)	0.19 (0.00, 0.45)	<0.001
H6P, µmol/L	0.64 (0.26, 1.40)	0.60 (0.25, 1.33)	0.65 (0.26, 1.44)	0.68 (0.28, 1.44)	0.56 (0.22, 1.20)	0.021
H7P, µmol/L	0.34 (0.13, 0.64)	0.32 (0.12, 0.58)	0.33 (0.14, 0.64)	0.37 (0.14, 0.73)	0.36 (0.14, 0.69)	<0.001

Continuous variables are reported as mean (SD) or median (interquartile range) and categorical variables are reported as n (%). Abbreviations: ALT, alanine aminotransferase; AST, aspartate aminotransferase; CRP, C-reactive protein; GFR, glomerular filtration rate (as calculated using the Chronic Kidney Disease Epidemiology Collaboration combined creatinine–cystatin C equation); GGT, gamma glutamyltransferase; H1P–H7P, high-density lipoprotein 1–7 particles (from smallest to largest); HDL-C, high-density lipoprotein cholesterol; HDL-P, high-density lipoprotein particles; IQR, interquartile range; SD, standard deviation. Small HDL: H1P and H2P; medium HDL: H3P–H4P; large HDL: H5P–H7P.

**Table 2 ijms-25-02290-t002:** Multivariable linear regression analyses showing relationships of HDL particles and subspecies with alcohol consumption.

Alcohol Consumption, g/Day	Beta (SE)	*p*-Value
**HDL-C, mmol/L**		
0/rarely	ref	
0.1–10	0.074 (0.009)	<0.001
10–30	0.174 (0.011)	<0.001
>30	0.270 (0.023)	<0.001
**HDL-P, μmol/L**		
0/rarely	ref	
0.1–10	0.585 (0.085)	<0.001
10–30	1.442 (0.107)	<0.001
>30	2.146 (0.202)	<0.001
**Large HDL, μmol/L**		
0/rarely	ref	
0.1–10	0.201 (0.040)	<0.001
10–30	0.415 (0.049)	<0.001
>30	0.566 (0.085)	<0.001
**Medium HDL, μmol/L**		
0/rarely	ref	
0.1–10	0.341 (0.066)	<0.001
10–30	1.009 (0.086)	<0.001
>30	1.857 (0.190)	<0.001
**Small HDL, μmol/L**		
0/rarely	ref	
0.1–10	0.044 (0.087)	0.61
10–30	0.018 (0.110)	0.87
>30	−0.277 (0.214)	0.19
**HDL size, nm**		
0/rarely	ref	
0.1–10	0.070 (0.013)	<0.001
10–30	0.156 (0.016)	<0.001
>30	0.259 (0.029)	<0.001
**H1P, μmol/L**		
0/rarely	ref	
0.1–10	−0.038 (0.058)	0.51
10–30	−0.228 (0.070)	0.001
>30	−0.705 (0.135)	<0.001
**H2P, μmol/L**		
0/rarely	ref	
0.1–10	0.082 (0.077)	0.29
10–30	0.247 (0.095)	0.009
>30	0.428 (0.183)	0.020
**H3P, μmol/L**		
0/rarely	ref	
0.1–10	0.190 (0.058)	0.001
10–30	0.623 (0.073)	<0.001
>30	1.119 (0.165)	<0.001
**H4P, μmol/L**		
0/rarely	ref	
0.1–10	0.151 (0.035)	<0.001
10–30	0.386 (0.045)	<0.001
>30	0.737 (0.092)	<0.001
**H5P, μmol/L**		
0/rarely	ref	
0.1–10	−0.017 (0.016)	0.27
10–30	−0.062 (0.018)	0.001
>30	−0.103 (0.028)	<0.001
**H6P, μmol/L**		
0/rarely	ref	
0.1–10	0.143 (0.031)	<0.001
10–30	0.291 (0.038)	<0.001
>30	0.370 (0.058)	<0.001
**H7P, μmol/L**		
0/rarely	ref	
0.1–10	0.075 (0.014)	<0.001
10–30	0.185 (0.019)	<0.001
>30	0.299 (0.038)	<0.001

Betas are unstandardized regression coefficients. SE, standard error; H1P–H7P, high-density lipoprotein 1–7 particles (from smallest to largest); HDL-C, high-density lipoprotein cholesterol; HDL-P, high-density lipoprotein particles; ref, denotes the reference category used for comparison; regression analyses were adjusted for age, sex, smoking status, history of type 2 diabetes, systolic blood pressure, total cholesterol, albumin, gamma glutamyltransferase, and alanine aminotransferase.

**Table 5 ijms-25-02290-t005:** Risk discrimination upon addition of HDL parameters to the WHO CVD risk prediction model containing conventional risk factors.

HDL Parameters	C-Index (95% CI): Conventional Risk Factors	C-Index (95% CI): Conventional Risk Factors Plus HDL Parameter	C-Index Change (95% CI); *p*-Value
HDL-C	0.7932 (0.7749 to 0.8116)	0.8028 (0.7846 to 0.8209)	0.0096 (0.0020 to 0.0171); *p* = 0.013
HDL-P	0.7932 (0.7749 to 0.8116)	0.7970 (0.7784 to 0.8157)	0.0038 (−0.0019 to 0.0065); *p* = 0.19
Large HDL	0.7932 (0.7749 to 0.8116)	0.7975 (0.7792 to 0.8158)	0.0043 (−0.0011 to 0.0097); *p* = 0.12
Medium HDL	0.7932 (0.7749 to 0.8116)	0.7962 (0.7776 to 0.8149)	0.0030 (−0.0025 to 0.0086); *p* = 0.29
Small HDL	0.7932 (0.7749 to 0.8116)	0.7933 (0.7749 to 0.8116)	0.0000 (−0.0008 to 0.0009); *p* = 0.95
HDL size	0.7932 (0.7749 to 0.8116)	0.7996 (0.7816 to 0.8176)	0.0064 (0.0004 to 0.0124); *p* = 0.038
H1P	0.7932 (0.7749 to 0.8116)	0.7932 (0.7748 to 0.8155)	−0.0000 (−0.0004 to 0.0003); *p* = 0.85
H2P	0.7932 (0.7749 to 0.8116)	0.7933 (0.7750 to 0.8116)	0.0001 (−0.0006 to 0.0008); *p* = 0.85
H3P	0.7932 (0.7749 to 0.8116)	0.7939 (0.7753 to 0.8125)	0.0007 (−0.0025 to 0.0038); *p* = 0.69
H4P	0.7932 (0.7749 to 0.8116)	0.7983 (0.7802 to 0.8164)	0.0051 (0.0000 to 0.0101); *p* = 0.050
H5P	0.7932 (0.7749 to 0.8116)	0.7934 (0.7751 to 0.8117)	0.0002 (−0.0005 to 0.0009); *p* = 0.61
H6P	0.7932 (0.7749 to 0.8116)	0.7972 (0.7789 to 0.8155)	0.0040 (−0.0018 to 0.0097); *p* = 0.18
H7P	0.7932 (0.7749 to 0.8116)	0.7981 (0.7799 to 0.8163)	0.0049 (−0.0008 to 0.0105); *p* = 0.091

The model with conventional risk factors included age, smoking status, systolic blood pressure, history of diabetes, and total cholesterol; CI, confidence interval; CVD, cardiovascular disease; H1P–H7P, high-density lipoprotein 1–7 particles (from smallest to largest); HDL-C, high-density lipoprotein cholesterol; HDL-P, high-density lipoprotein particles; WHO, World Health Organization.

## Data Availability

The data that support the findings of this study are available on reasonable request from the Principal Investigators.

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
