# Peer review of "Alcohol Consumption, High-Density Lipoprotein Particles and Subspecies, and Risk of Cardiovascular Disease: Findings from the PREVEND Prospective Study"

_ijms, 2024, doi:10.3390/ijms25042290_

Round 1

Reviewer 1 Report

Comments and Suggestions for Authors

Abstract

This is an interesting approach on HDL-C. This article aims to analyze the complex and controversial interaction between alcohol consumption, HDL cholesterol (HDL-C) parameters, and its sub-particles, and cardiovascular risk. The participants belonged to the PREVEND cohort (5,309 Dutch participants), and the STROBE method for observational epidemiological studies was used. Sociodemographic, lifestyle, anthropometric, medical history, medication use, and self-reported alcohol consumption data were collected; in addition, a blood sample was obtained. The main objectives of this study were to investigate the association of HDL-C, as well as its sub-particles, with self-reported alcohol consumption, and secondly to analyze the effect of various HDL-C parameters on cardiovascular risk, considering alcohol consumption as a confounder and effect modifier.

The following are some comments and suggestions per section to be considered by the authors. 

Introduction

1) The introduction only emphasizes studies in which a beneficial effect of alcohol consumption on health has been found; it would also be important to mention the controversial information that exists in this regard, since due to the nutritional characteristics of alcoholic beverages it is not possible to implement a dietary recommendation for their consumption.  

2) When mentioning the controversial role of HDL-C in cardiovascular risk, it is suggested to mention which sub-particles have been related to a better or worse prognosis of cardiovascular diseases, according to their size, composition, and so forth.

Materials and Methods

1) Self-reporting may generate bias in reporting data. Was any method used to reduce bias? If so, what was it?

2) Although an alcoholic beverage corresponds to 10 g of alcohol, it is important to delimit if all alcoholic beverages, whether distilled, beer or cocktail, have the same amount of alcohol.

3) It is suggested to mention the source from which the measurement values were taken for each of the HDL-C sub-particles.

Results

1)    These may be divided for men and women and be used as well to detect a cardiovascular risk according to WHO.

2)    Why were no hepatic biomarkers considered, such as γ-glutamyltransferase (GGT) activity, aspartate aminotransferase (AST) activity?

3)    Make sure that the Figure 1 is in high quality resolution, It seems to be blurry. It may be more attractive to use a heat map instead and mark the significant values. 

4)    Be consistent in placing periods at the end.

5)    It would be suggested that a different type of chart may be used to provide a more visual information regarding the associations.

6)    The results address cardiovascular disease although a link towards a cardiovascular risk is not fully addressed.

Discussion

1) It is controversial to emit a recommendation on alcohol consumption, especially in the context of cardiovascular risk. It is suggested to compare with other articles in which the association is negative and evaluating the risk-benefit of the recommendation. In addition, the characteristics of alcoholic beverages make them of low nutritional value, so it is not recommended to include them in a healthy diet.

2) The proposed molecular mechanisms are interesting, but further analysis is recommended.

3) The studies cited are from 2001 and 2011, it is advisable to look for more updated information for discussion.

4) In order to have a broader perspective, you could analyze articles in which lipid profile molecules such as LDL-C, VLDL-C or triglycerides are related to alcohol consumption. 

5) These associations remained unmodified by additional adjustment for alcohol consumption.

6) It would be needed to correct the word Dallas (line 277) and HDL-P (line 294).

7) In line 295 “in” and “with” may be deleted.

8) The array of HDL is something to stand out as you don´t see this often in other articles.

Reviewer 2 Report

Comments and Suggestions for Authors

In the PREVEND study, a large general population-based prospective study of 5,309 subjects, the authors assessed self-reported alcohol intake and HDL-C and HDL-P and subspecies (small, medium, and large) by nuclear magnetic resonance spectroscopy. Increased alcohol intake is associated with increased HDL-C, HDL-P, large and medium HDL, and HDL size; with respect to CVD, only HDL-C was associated with a statistically significant reduction in CVD risk in fully adjusted analyses and is largely independent of HDL parameters.

This paper is very novel and interesting in that it examines the relationship between alcohol intake, HDL-C, and HDL size by nuclear magnetic resonance spectroscopy, and evaluates the relationship between alcohol intake and CVD.

However, several problems are noted.

The authors evaluate the effects of alcohol consumption on HDL cholesterol levels and HDL size by NMR, but do not evaluate the functionality of HDL. In recent years, the importance of HDL research has focused on various functions of HDL particles, including their cholesterol efflux capacity or antioxidation function. In addition to the effect of alcohol consumption on HDL cholesterol content and HDL particle size, the effect on HDL functionality should also be shown. At the very least, there should be a statement in the Discussion regarding the association between alcohol intake and HDL function.

As the authors note in the Discussion, previous studies have already suggested an association between alcohol intake and CETP and apolipoprotein A-I. This study should show various apolipoproteins and CETP activity or CETP mass.

Heavy drinkers often have liver damage and poor nutritional status. Therefore, baseline characteristics should include liver function such as AST/ALT/γGTP and platelets and total protein/albumin.

In the discussion of this paper, it is described that HDL-P is more associated with myocardial infarction than HDL-C. The authors should be more careful in describing the possible differences between HDL-P and HDL-C, as this is relevant to the core of this study.

Round 2

Reviewer 1 Report

Comments and Suggestions for Authors

Introduction 

Thank you for addressing previous comments, there are no further comments in this section.

Material and methods

I am grateful for the explanation about the possible bias caused by self-reporting and, above all, that it is added as a limitation of the study, as well as for the comment on the standardization of alcoholic beverages to 10 g of alcohol; these methodological aspects will be of help for future research on the same topic. No further comments in this section

Results

  1. Regarding Figure 1. Correlation heat map and matrix for HDL parameters. Here as a suggestion to be easier for the reader to understand you may include the correlation values instead of having two figures and highlight the significant values.

Discussion

  1. in line 372 “Dallas” has yet to be written correctly

The discussion section was significantly strengthened by expanding the molecular mechanisms, including other molecules of the lipid profile, as well as updating the articles that are included.

Reviewer 2 Report

Comments and Suggestions for Authors

The authors have carefully revised the paper to the extent possible in response to the reviewers' comments. The authors are to be commended for their efforts. We consider the paper to be of very high quality.

No further comments.
